# Neuro-Symbolic Representations of 3D Scenes Using Universal Scene Description Language

**M Shifat Hossain[1,*], Md Rubel Ahmed[1,*], Laura Pullum[2], Sumit Jha[3], Rickard Ewetz[1]**

[1]University of Central Florida, Orlando, Florida, USA
[2]The POM Group, Oak Ridge, Tennessee, USA
[3]Florida International University, Miami, Florida, USA
mshifat.hossain@ucf.edu, mdrubel.ahmed@ucf.edu, laurapullum@gmail.com, jha@cs.fiu.edu, rickard.ewetz@ucf.edu

## Abstract

Developing efficient and expressive representations of 3D scenes is a pivotal problem within 3D computer vision. The state-of-the-art approach is based on utilizing 3D point clouds, which is inefficient in data utilization. In this paper, we propose a neuro-symbolic approach leveraging the Universal Scene Description (USD) language. The approach is based on representing 3D scenes using a combination of known objects (symbolic) and 3D point clouds (neural) for the background. We also propose a framework called neuro-symbolic conversion (NSC) for automatically converting 3D scenes into the proposed neuro-symbolic representation. The NSC framework first locates candidate objects in the 3D point cloud representation. Next, the objects are substituted with their compact symbolic representation while considering translations and rotations. The correctness of the substitution is verified by rendering the neuro-symbolic representation and comparing the visual similarity with the original point cloud representation (or RGB-D view). The experimental results demonstrate that our framework is highly accurate in object identification and objection substitution. The neuro-symbolic representations are expected to be useful for downstream tasks such as entity identification, activity recognition, and object tracking.

## 1 Introduction

3D computer vision is vital in machine learning research, significantly contributing to spatial perception and automation, especially in manufacturing, healthcare, and defense. Recent advancements in 3D computer vision include enhanced semantic segmentation (Barbosa and Osório 2023), robust object pose estimation techniques (Zhu et al. 2022), and the integration of 3D vision into autonomous systems (Singh and Bankiti 2023). Traditional explicit 3D representations, like point clouds, meshes, and voxels, encounter challenges in efficiently handling complex and deformable shapes, facing issues in accurately capturing details and managing incomplete or noisy data. Point clouds, utilized for 3D data capture, confront challenges, notably in quantization, leading to precision loss, and their unstructured nature creates compatibility issues with traditional machine learning techniques. Furthermore, innovative methods

---

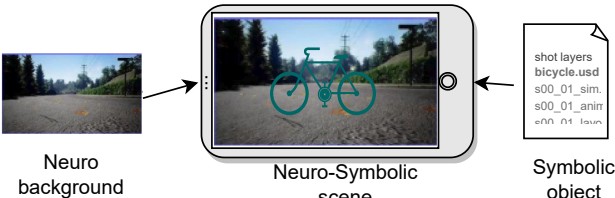

Figure 1: (left) Neural representation of background using 3D point cloud. (right) Symbolic representation of a known object in universal scene description (USD) language. (middle) Neuro-symbolic representation of a 3D scene.

like voxel-based CNNs (Liu et al. 2019) and graph-based approaches (Mirande et al. 2022) introduce complexities and high computational overhead when transforming point clouds into structured formats.

Symbolic representations of data offer computational efficiency and storage benefits, making them an attractive choice for low-dimensional data. However, their limitations become apparent in high-dimensional scenarios where symbolic approaches may lack expressiveness to effectively capture intricate patterns, presenting challenges in preserving the rich information associated with complex datasets. USD, embraced in production houses like DreamWorks (Blevins and Murray 2018), becomes an industry standard due to its versatility and user-friendly features.

In this paper, we propose a neuro-symbolic approach for representing 3D scenes using USD. The approach integrates known objects (symbolic) with 3D point clouds (neural), resulting in compact and efficient neuro-symbolic representations of intricate 3D scenes, as illustrated in Figure 1. We also propose a framework called neuro-symbolic conversion (NSC) for converting RGB-D images (or 3D point clouds) into the proposed neuro-symbolic representation using USD. The USD format can render back the original RGB-D image. This bidirectional conversion adeptly represents complex 3D scenes. It demonstrates practical applications by seamlessly substituting objects from a library into the USD format. This framework has the potential for efficient scene manipulation and object recognition. To the best of our knowledge, our neuro-symbolic framework represents a pioneering exploration of USD, offering more efficient and adaptable object representations. Our main **contributions** can be summarized as follows:

- A neuro-symbolic representation of 3D scenes in USD format. The format supports bidirectional conversions between RGB-D images and neuro-symbolic 3D scenes.
- Introducing an effective approach called neuro-symbolic conversion (NSC) for identifying and matching objects within a 3D scene using a predefined object library.
- The experimental results show that NSC can identify objects with 100% accuracy and substitute each object with more than 90% similarity on average.

The organization of this paper is as follows. The remainder of this section reviews some closely related works. Section 2 formulates the problem, and Section 3 describes the proposed 3D scene to USD object conversion and reproduction method. Section 4 presents the experimental results, and the last section concludes the paper.

**Previous Works.** Efficiently representing 3D data is crucial with the growing use of 3D technologies in applications such as virtual reality, mobile mapping, historical artifact scanning, and 3D visualization (Sugimoto et al. 2017; Wandersman 2023). The work (Nguyen et al. 2023) investigates virtual reality technology for robot environment modeling and presents a method to translate USD-based scene graphs into Knowledge Graphs (KGs). The resulting KG, augmented with dynamic data from a physics simulator, acts as background knowledge for robotic decision-making, demonstrated in a box unpacking scenario. Despite these advances, exploring deep learning with symbolic representations, like USD, remains relatively uncharted. This research avenue holds the promise of seamlessly combining the advantages of compact 3D scene representation in efficient USD data structure, offering potential benefits for machine learning models to operate more efficiently.

Learning in a 3D environment is an active research area. The unidirectional transformer-based approach (Hong et al. 2023) presents a Large Reconstruction Model, capable of rapidly predicting 3D object models from single input images, trained on an extensive multi-view dataset for enhanced generalizability and performance across various testing scenarios. Neuralangelo (Li et al. 2023) combines multi-resolution 3D hash grids with neural rendering, utilizing numerical gradients and a coarse-to-fine optimization strategy to achieve superior 3D surface reconstruction from multi-view images. Our research focuses on utilizing USD representations of 3D objects to enhance downstream machine learning applications within the 3D environment.

## 2 Neuro-Symbolic Representations of 3D Scenes using USD

Conversion from a neural representation to a neuro-symbolic representation requires substituting parts of neural representations with symbolic representations. This substitution requires identifying objects of interest from the neural representation and extracting essential information related to the objects in question. The information extracted from the neural images can then be used to reconstruct a symbolic representation of each object and can then be replaced, constructing a neuro-symbolic representation. These neuro-symbolic representations stored in USD formats can be used to store, interpret, analyze, and view neuro-symbolic data.

### 2.1 Problem Formulation

Finding the solution to this complex problem can be formulated using the following optimization function:

$$\max_{\theta} SSIM(I, J_{\theta}) \tag{1}$$

Here, the terms $I$ and $J_{\theta}$ are the neural image and neuro-symbolic images with different parameters $\theta$, respectively. The neuro-symbolic images are generated based on the detection of an object of interest from the neural image, and then parameters ($\theta$) of the symbolic objects are optimized by maximizing the Structural Similarity Index (SSIM) score.

## 3 Overview of Proposed Methodology

In this section, we provide the details of the neuro-symbolic conversion (NSC) framework. The flow of the framework is shown in Figure 2 and illustrated with an example in Figure 3. The input to the framework is an RGB-D image. The output is a neuro-symbolic representation in USD. The first step of the framework is to construct a 3D point cloud (neural) representation of the scene. The next step is to identify and substitute portions of the image with objects of interest (symbolic) from a library. The substitution is performed such that a perception-based metric is minimized.

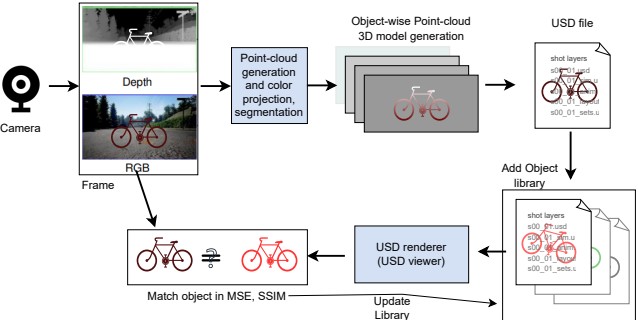

Figure 2: Overall architecture of proposed NSC framework explaining proposed neural and neuro-symbolic bidirectional conversion procedure.

### 3.1 Bidirectional Conversion

The process of seamlessly converting images to and from USD involves harnessing the high-performance capabilities of the USD software platform. In the forward process, RGB and depth images are rendered from existing USD files. This bidirectional conversion process ensures a holistic transformation between neural representations and symbolic descriptions. According to the USD 1.0 specification, a USD can contain different primitive objects. These primitive objects are the nodes that store the mesh and other objects (e.g., lights, cameras, etc.) of a scene. These USD files can then be rendered using a USD rendering engine. We utilize "PointInstancer" to store the neural point cloud data and primitive types to store symbolic library objects in NSC.

Conversely, in the reverse process, USD files are generated from RGB-depth images. For instance, RGB and depth

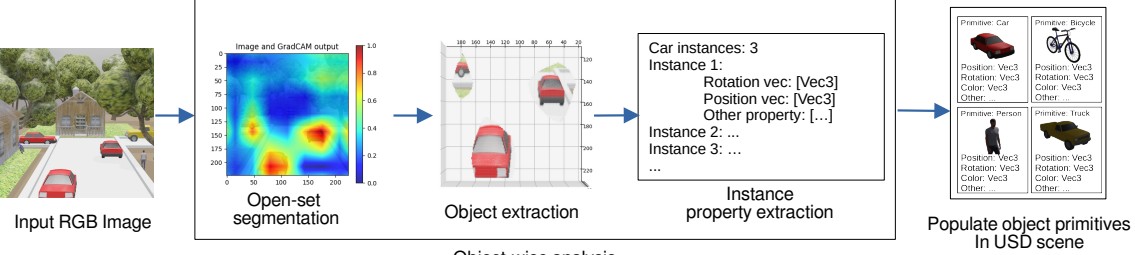

Figure 3: Semantic segmentation guides the extraction of object point clouds, analyzed for primitive object properties ($Params_i$). These properties drive the placement of primitives in the scene, facilitating USD reconstruction.

frames captured from DARPA Airsim are utilized in our specific implementation to form the basis for USD scene generation. This intricate conversion process is performed using USD-core and Kaolin libraries.

## 3.2 Image to Object-wise USD Scene Generation

This step involves semantic segmentation, point-cloud generation, object selection and identification, reconstruction, and USD export. Consider the capture of RGB color images, denoted as $I_{RGB}$, and depth images, denoted as $I_{Depth}$, from the Airsim environment. To semantically segment the color images, a proposed open-set semantic segmenter is applied, producing semantic segmentation maps denoted as $M_{Semantic}$. Concurrently, the depth images are utilized to generate point-cloud information represented as $P_{Cloud}$.

The semantic segmentation maps $M_{Semantic}$ are then intricately projected onto the point-cloud data $P_{Cloud}$. This projection serves the purpose of symbolically identifying distinct objects within the 3D environment. Mathematically, this projection operation can be expressed as.

$$O_{Symbolic} = \text{Project}(M_{Semantic}, P_{Cloud}) \quad (2)$$

Here, $O_{Symbolic}$ is the set of symbolically identified objects from the scene. Subsequently, individual objects are selected from the symbolic identification $O_{Symbolic}$ and undergo direct USD export or reconstruction from the point-cloud information to form 3D meshes. Let $M_i$ denote the mesh of the $i^{th}$ object, and $O_i$ represent the $i^{th}$ selected object. The reconstruction operation can be represented as: $M_i = \text{Reconstruct}(O_i, P_{Cloud})$.

## 3.3 Image to Object-Wise Property Extraction

The object property extraction from images involves projecting a semantic segmentation map $M_{Semantic}$ to identify object point clouds $P_{Object}$ as shown in Figure 3. The object-wise point clouds are then analyzed to extract parameters $\theta = \{location, rotation, color, \dots\}$ characterizing each primitive object. These parameters are utilized to place primitives in the scene. Finally, a USD scene is reconstructed by optimizing the parameter set $\theta$ using equation 1, combining primitives and their extracted properties. The objective of this step is to ensure the effective extraction and utilization of object properties within the USD environment.

The proposed method NSC is outlined in Algorithm 1. The goal is to find the best match between a test image and library objects. Using GradCAM (Selvaraju et al. 2017),

---

**Algorithm 1: Neuro-Symbolic Conversion**

**Require:** Neural image $I_{test}$, SymbolicObjectLibrary(SOL)
**Ensure:** Neuro-Symbolic representation of 3D image
1: **function** MATCHOBJECTS($I_{test}$, SOL)
2:     **for** each object of interest $O_i$ in SOL **do**
3:         GradCAM identifies point cloud of $O_i$: $P_{object}$
4:         **for** each identified point **do**
5:             Remove pixel: $I_{modified} \leftarrow I_{test} \setminus P_{object}$
6:             **for** each object $\theta$ set **do**
7:                 Compute SSIM and MSE scores
8:                 Track scores for each variation
9:             **end for**
10:             Select best-matched variation $O_t$
11:             Replace removed pixel with $O_t$
12:         **end for**
13:     **end for**
14:     **return** $I_{modified} + O_t$
15: **end function**

---

the algorithm identifies the pixels in the test image corresponding to the objects of interest. Subsequently, it performs pixel removal and replacement with various library object variations, considering different orientations and colors. The matching process involves computing SSIM and Mean Squared Error (MSE) scores between the modified test image and rendered images with library object variations. The algorithm iterates through possible variations, keeping track of scores for each. The best match for each object is one that has maximum SSIM and minimum MSE scores. The outcome is a neuro-symbolic image showcasing the best-matched variations of the objects of interest. This approach enables efficient estimation of categorical and transformational information for important objects in the scene.

## 4 Experimental Results

In this section, we validate NSC through two distinct 3D scenarios involving five different scenes featuring a red car and a human as our objects of interest. The objective is to apply NSC to identify point clouds representing these objects, subsequently substituting them with corresponding symbolic library objects. The modified scenes are then compared with the ground truth scenarios on SSIM and MSE metrics to find the best match. Upon finding a highly matching object, transformation information is extracted and stored in USD.

Illustrated in Figure 4, NSC demonstrates notable effectiveness in substituting objects within 3D scenes. The red

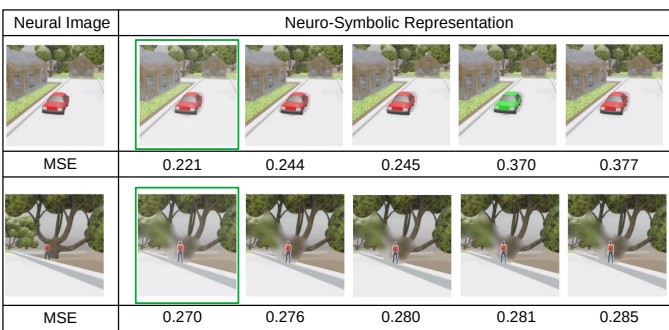

| Neural Image | Neuro-Symbolic Representation | | | | |
|---|---|---|---|---|---|
| MSE | 0.221 | 0.244 | 0.245 | 0.370 | 0.377 |
| MSE | 0.270 | 0.276 | 0.280 | 0.281 | 0.285 |

Figure 4: Best matching scene selection based on the MSE. Objects of interest are car (top) and human (bottom).

Table 1: Object (**Obj.**) property extraction.

| Scene | Obj. | SSIM | Accur. | Err. $|\theta^{rad}|$ | Err. $|Y|$ |
|---|---|---|---|---|---|
| 1 | Car | **0.98** | 100% | 0.003 | 0 |
| 2 | Human | **0.92** | 100% | 0.012 | 0 |
| 3 | Car | 0.97 | 100% | 0.003 | 0 |
| 4 | Human | 0.91 | 100% | 0.012 | 0 |
| 5 | Car | 0.97 | 100% | 0.004 | 0 |
| Average | | 0.95 | 100% | 0.006 | 0 |

car, enclosed in a green box, exhibits the lowest normalized MSE, indicating the highest match with the ground truth neuro-image. MSE is normalized by the image size ($224 \times 224$) and multiplied by 100 for better representation. Figure 4 also showcases the top five USD representations out of 108 car object variants compared in this scene. Table 1 reports property extractions for the corresponding car object. A high SSIM value, along with very low errors (**Err.**) in rotation (in radians) and relative depth (in meter) as shown in **Scene** 1, instills confidence in the identified objects. The scene can be further annotated with the library objects' known attributes, facilitating numerous downstream learning and analysis tasks efficiently. The **Accur.** (accuracy) column highlights that object identification is correct for all instances (108 out of 108) in this example scenario.

A similar trend is observed in MSE and SSIM scores for the human, albeit not as ideal as the car object. Humans are correctly identified in all scenes (72 out of 72) and replaced accurately. Our investigation reveals that the human object's location in the scene relative to the surroundings contributes to poorer MSE and SSIM compared to the car object.

NSC iteratively estimates both categorical and transformational information for essential objects, facilitating the update of object attributes within the USD library. The preliminary results affirm the feasibility of NSC in intricate neuro-symbolic AI tasks.

## 5 Conclusion and Future Work

Our work demonstrates a substantial advancement in the neuro-symbolic representation of 3D scenes. The bidirectional image-to-USD conversion, object-wise property extraction, and dynamic library matching highlight the efficiency and adaptability of our approach. In future, we plan to expand our method to enhance the symbolic reasoning of machine learning models, with a specific focus on deeper exploration of inter-symbolic relationships.

**Acknowledgment.** The authors were in part supported by DARPA Co-operative Agreements #HR00112020002, #HR00112420004, and #FA8750-23-2-0501, and DOE grants #DE-SC0023494 and #DE-SC0024576. The views, opinions and/or findings expressed are those of the authors and should not be interpreted as representing the official views or policies of those providing support for this work.

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
