# OpenReview forum: "Neuro-Symbolic Representations of 3D Scenes using Universal Scene Description Language"
_AAAI.org/2024/Workshop/NuCLeaR — NuCLeaR 2024_

### Official Review · Reviewer_18n5 · 2023-11-29
**Neuro-Symbolic Representations of 3D Scenes using Universal Scene Description Language**

**Rating:** 7
**Confidence:** 3

**Review:**

This work is well written though it could have presented more information on the approach. And could have done a better comparison against other similar approaches to show any sort of improvement over competing approaches. The question why is this approach better than the current state of the art was never quite answered. Just because you can do something doesn't mean you should do it.

Neural Radiance Fields or other modern representations could also be mentioned for comparison, since point clouds have been improved upon.

I don't know what their page count limit was, but the paper could be extended to six or eight pages to show the USD representation and the final neuro-symbolic representation of a single processed image, to further highlight what specifically is gained by taking this approach.  Also the reduction in space could be discussed, i.e., how much space is required by the original point cloud representation versus how much space is required by the final neuro-symbolic / USD representation.

What was glaringly absent from this discussion was the amount of time taken to process a single image. No timing information was presented at all, which is concerning.

The fact that a library of object images needs to be pre-constructed as well is also concerning. It is not mentioned where this library comes from.

For the [near/far] future, it would be nice to generate the matched object library directly from the scene itself. When the object library becomes enormous, how costly will it be to match extant known objects against a new scene that is being processed in real-time, (24 frames per second).   Obviously the matching should be sub-second.

Despite these shortcomings of the current paper, I think that this investigation is certainly noteworthy.

The paper should be revised to address the shortcomings mentioned above.

---

### Official Review · Reviewer_cuno · 2023-12-08
**A Review of NuCLear Submission 21**

**Rating:** 5
**Confidence:** 3

**Review:**

The authors have convinced me that USD is worth investigating for data storage, when AI/ML operations over 3DPC images is being done.  Their "NSC" is apparently a useful way to convert between 3DPC and NS representations.  No concern is given to processing time, suggesting that the ground-truth-preserving and cost-optimized results of NSC are of high enough value in themselves to ignore such overhead initially.  The same is true for the particulars of the NSC method--they are not inevitable, and have cost, but the fact that the NSC includes iterations through an object library, allowing for transformations, seems to make NSC's versatility more prominent than its speed.


What I don't know yet:
* what, if any, selection procedure was used to yield the particular semantic segmenter they used, and/or its invocation parameters?
* what data storage savings are realized by using NS representations (in USD, in this case)?  I can easily infer that they must exist, but since the authors explicitly state that 3DPC representations are inefficient, I'd like to see a contrast.
* how does the computational overhead of using NSC's central algorithm compare with the computational overheads associated with voxel-based CNNs and graph-based approaches?
* what is it about USD that makes it, as opposed to other storage formats, especially suitable for this purpose?  Again, I can see that it probably is (due to extensibility and ecosystem), but all we're told is that USD is considered versatile and user-friendly in "production houses", and has "high-performance capabilities".  I infer that "permits use of things like pointinstancer" is one such capability, but again not explained.
* what drove their selection of USD rendering engine?  Which one was it?  If it didn't matter which one, I'd like to see that asserted.
* what example AI/ML research workflow would this be a part of?

There are places where the paper's text flows poorly (below).  More importantly, while length limit can explain the persistence of my "unknowns" above, I think that (e.g.) USD could have been given more of an introduction at the sentence or paragraph level.  Also wanted to see some elaboration on the nature or quantity of the space/expressiveness "efficiency gains" that NS or USD representations offer.  Both of these things even within a space constraint.

If the paper had addressed those things, or had gone for the 8-page length and addressed items like my "remaining unknowns" more, and/or otherwise usefully lengthened the paper, I'd consider it an 'above threshold'.

Suggested text replacements (there might be another place or two where a pronoun doesn't match its referent in number):

Page 1:

* This lacks a verb:  "Recent advancements in 3D computer vision, including enhanced semantic segmentation (Barbosa and Os´orio 2023), robust object pose estimation techniques (Zhu et al. 2022), and the integration of 3D vision into autonomous systems (Singh and Bankiti 2023)."
* "Symbolic representations of data offer computational efficiency and storage benefits, making it an attractive choice for low-dimensional data. However, its limitations"
'representations' is plural, so should say 'making them an attractive choice' and 'their limitations'.
* "USD is embraced..."  USD is not defined here, so I can tell it has nice features, but not what it is.

Page 2:

* (BTW Section 2 title contains the string "Scens using USD".  It should have "Scenes".)

* "requires substituting part" should say "requires substituting parts".

* "related to the object in question" should say "related to the objects in question".

* "reconstruct symbolic representation of the object" should say "reconstruct symbolic representations of each object".

* "This research avenue holds the promise of seamlessly combining the advantages of compact 3D scene representation in this efficient data structure than point cloud."  The word "than" leads me to look for two things being compared, but I don't see that happening here.

* "These neuro-symbolic hybrid representations stored in USD formats..."  I think this is referring to data stored using the 'pointinstancer' schema.  The term 'hybrid' appears nowhere else in the paper--I'd like it to do so, or else not appear here.

---

### Decision · Program_Chairs · 2023-12-11

**Decision:**

Accept

**Comment:**

I have changed the order of the authors.